# Nanoplastics and Human Health: Hazard Identification and Biointerface

**DOI:** 10.3390/nano12081298

**Published:** 2022-04-11

**Authors:** Hanpeng Lai, Xing Liu, Man Qu

**Affiliations:** School of Nursing & School of Public Health, Yangzhou University, Yangzhou 225000, China; alfred_lai@126.com (H.L.); liuxing@yzu.edu.cn (X.L.)

**Keywords:** nanoplastics, human health, hazard identification, biointerface

## Abstract

Nanoplastics are associated with several risks to the ecology and toxicity to humans. Nanoplastics are synthetic polymers with dimensions ranging from 1 nm to 1 μm. They are directly released to the environment or secondarily derived from plastic disintegration in the environment. Nanoplastics are widely detected in environmental samples and the food chain; therefore, their potentially toxic effects have been widely explored. In the present review, an overview of another two potential sources of nanoplastics, exposure routes to illustrate hazard identification of nanoplastics, cell internalization, and effects on intracellular target organelles are presented. In addition, challenges on the study of nanoplastics and future research areas are summarized. This paper also summarizes some approaches to eliminate or minimize the levels of nanoplastics to ensure environmental safety and improve human health.

## 1. Introduction

Microplastics were initially reported in 2004, and the potential adverse effects on living organisms have recently attracted considerable attention [1,2]. Microplastics are defined as artificial polymer particles with size less than or equal to 5 mm. They can be produced through structural disintegration of waste plastic products and released to the environment [3,4,5]. Notably, microplastic particles are not the end products of plastic waste as they decompose into nanoplastics [6,7]. Nanoplastics are defined as particles with a size ranging between 1 nm and 1 μm, which is different from the classification criteria of engineered nanomaterials (ENMs) [8,9,10]. The bioeffects of nanoplastics should be fully explored owing to their potentially enhanced toxicity and small size.

Plastic objects are commonly available in every aspect of our life, such food containers, tap-water pipes, and the clothing industry [11,12,13]. Plastic particles in microscale or nanoscale are used in drug delivery, personal care products, and bioimaging [14,15,16]. Several studies indicate that water bodies, air, soil, food, and table salt may contain microplastic or nanoplastic particles [17,18,19,20]. However, studies have not fully elucidated the effect of nanoplastics on human health. Therefore, the negative impacts of nanoplastics on the health of human beings should be explored.

In the following parts, we present the major sources, exposure routes, biointerface, and intracellular target organelles of nanoplastics. Beyond that, some directions of future research to better comprehend or solve nanoplastic issues have simultaneously been suggested.

## 2. Another Two Sources of Nanoplastics in the Environment

Microplastics or nanoplastics mainly originate from two major sources, namely, primary plastic waste and secondary derivatives [21,22]. Primary microplastic or nanoplastic wastes are disposed to the environment in microscale or nanoscale form, including sources such as nanomedicine, nanoimaging, nanosensors, and personal care products [14,16,23]. Secondary derivatives are microplastics or nanoplastics derived from disintegration of plastics driven by physical, chemical, or biological forces [2,21]. Considering that tire wear and laundry wastewater are both notorious sources of microplastics, they might also be sources of nanoplastics. Hence, another two main sources of nanoplastic secondary derivatives are presented in the subsequent sections (Figure 1).

### 2.1. Nanoplastics from Tire Wear

Tires rub quickly against the ground when cars are at high speed, leading to the release of particles to the environment. Approximately 30% of the weight of a tire is emitted to the environment from use to scrap materials [24]. Simulation studies show that microscale and nanoscale particles are derived from tires under constant friction [25,26,27]. In addition, an analysis of airborne particles near a road showed that the size of the particles ranged from 6 to 562 nm and from 30 to 60 nm when cars were braking [28]. A previous study analyzed the composition of microplastics obtained from the Charleston Harbor Estuary, and the findings showed that tire wear particles ranked second with a 17.1% proportion of all detected microplastics [29]. This potential source of plastics should be given attention due to the potential adverse effects of microplastics or nanoplastics and the large amounts emitted from tire wear.

### 2.2. Nanoplastics from Laundry Wastewater

Laundry wastewater is a major source of plastics. Plastic fibers are among the most frequently detected microplastic or nanoplastic particles from environmental samples, which mainly originate from household washing [30,31,32]. Acrylic, nylon, and polyester microfibers are released from synthetic textiles to the laundry wastewater, with an average of 7360 fibers/m^−2^/L^−1^ [32]. Moreover, microfibers appear in the first wash wastewater of polyester and cotton textiles with a size ranging between 2.1 × 10^5^ and 1.3 × 10^7^ [33]. The annual emission of microfibers from polyester and cotton from household laundry wastewater was projected to be 565,000 kg each year [33]. Studies should explore this source of plastics, owing to the high emission capacity.

## 3. Potential Exposure Routes of Nanoplastics and Adverse Effects on Humans

### 3.1. Potential Exposure Routes of Nanoplastics to Humans

Humans are subjected to long-term (almost whole-life) exposure to nanoplastics at low concentrations [34,35]. Nanoplastics are difficult to detect compared with microplastics; therefore, their effects are not widely explored [36]. It is challenging to eradicate nanoplastics even in the short term and under environmentally realistic concentrations once living organisms are exposed to the particles [37]. In the present review, potential exposure routes are summarized to explore the effects of nanoplastics on humans (Figure 2).

#### 3.1.1. Oral Intake

Oral intake is the most evaluated and common route through which humans are exposed to nanoplastics. This route leads to the continuous intake of nanoplastics into the body. Extensive usage of plastic components in drinking water networks and the potential leach of microplastics or nanoplastics from water pipes under long-term fluid force leads to high exposure of high amounts of plastics to the consumers of drinking water [38,39,40]. A previous study reported that humans consume hundreds of millions of nanoplastic particles from a single cup of beverage originating from teabag packaging [31]. The detection of nanoplastics in seafoods as well as table salts shows that humans are exposed to high levels of plastics [41]. Nanoplastics originally absorbed by organisms lower on the food chain are further ingested by organisms higher in the food chain and eventually by humans, which affects human health through bioaccumulation [42]. This implies that nanoplastics get into the systems of humans through drinking water and eating of materials exposed to nanoplastics. Studies should evaluate the level of nanoplastics in drinking water and food, and effective detection approaches should be developed.

#### 3.1.2. Air Inhalation

Volatile chemical toxicants (such as cyanide and benzenamine) and particulate matters (such as PM 2.5 and PM 10) mainly get into human systems through air inhalation. Nanoplastics are novel carriers for chemical detriments (such as airborne PM 2.5) and are widely distributed in the air [30,43]. A previous study reported that nanoplastics appear in atmospheric fallout, indicating that people may inhale nanoplastics from the air [3]. In addition, the wearing out of car tires generates microplastics or nanoplastics, which are released to the air around the street, becoming a potential source of inhalation exposure [29,44]. A study on private residences and public offices reported that indoor air contains higher concentration of microplastics or nanoplastics compared with the outdoor concentrations, which implies that humans are exposed to high amounts of plastic microparticles or nanoparticles [30].

#### 3.1.3. Dermal Exposure

Besides oral intake and air inhalation, people may be subjected to dermal exposure of nanoplastics. Nanotoxicology of nanoengineered materials indicates that nanoparticles (including nanoplastics) with a size less than 40 nm may enter the body through the epidermal barrier [45,46]. Dermal exposure mainly occurs when humans come into close contract with nanoplastics in the environment, such as taking a shower with water or using personal care products containing nanoplastics such as nanopolystyrene [16,35].

### 3.2. Potential Adverse Effects of Nanoplastics on Human

Nanoplastics invade living organisms, cross biological barriers, and are transferred to their offspring [37,47]. Nanoplastics present in living organisms are only excreted at very low concentrations, leading to accumulation [37]. Studies have not fully explored the adverse health effects of nanoplastics in humankind. The present findings are mainly from nanotoxicology studies using nanopolystyrene and are summarized below.

In vitro and in vivo studies indicated that microplastics or nanoplastics absorbed into the human body accumulate in the intestinal lumen, and some of these plastics particles are excreted through feces [48,49]. Findings from animal experiments indicate that nanoparticles distributed in intestinal lumen can penetrate the intestinal barrier and can be further translocated into blood vessels [50]. Alveolus is the part of the lung with a comparatively larger pore size for releasing oxygen to the blood and receiving carbon dioxide from the blood; thus, it forms as a vital part of the blood–air barrier [51]. This implies that nanoplastics inhaled into human lungs may penetrate the blood–air barrier and may be transported into the blood-circulating system.

Moreover, nanoplastics can cross the blood–brain barrier after intravascular injection and accumulate in the brain [52]. A study on nanoplastics obtained through the food chain showed that nanoplastics cross the blood–brain barrier, inducing brain damage in fish [53]. Furthermore, a study using an ex vivo human placental perfusion model of nanoplastics reported that nanoplastics can cross the placental barrier through passive diffusion [54]. These findings indicate that nanoplastics can penetrate important biological barriers (such as the intestinal barrier, blood–air barrier, blood–brain barrier, and placental barrier) and pose potential adverse effects to people.

## 4. Behavior of Nanoplastics

### 4.1. Interactions with Biological Media

Nanoplastics are packaged by biomacromolecules (such as proteins and polypeptides) once they come into contact with biological milieu, forming a complex known as “corona” [55,56]. Protein corona are divided into two types: “hard” and “soft”. The “hard” comprises adsorption of the nanoplastics into the inner layer of proteins, whereas the “soft” corona means that the particles are loosely adsorbed on the outer layer of proteins [57]. Protein “corona” induces an alteration on the properties of its adsorbents (such as size and biological activities) and significantly affects the physicochemical properties of nanoplastics (such as size, charge, and shape) [58]. Studies report that protein corona modulates internalization of microparticles or nanoparticles into the cell [58,59].

### 4.2. Interactions with Cell Membrane

#### 4.2.1. Cell Internalization

Cells internalize micro particles or nanoparticles through two main ways, namely, passive targeting and active targeting. Passive targeting is a type of transport whereby substances (small molecules/particles) are transported across a concentration gradient or are based on the potential difference of the substances inside and outside the plasma membrane without ATP consumption. Active targeting refers to the transport process whereby substances are transported against the concentration gradient and requires energy [60]. These two transport modes exist concurrently in organisms for transport of nanoplastics absorbed from the environment.

Nanoplastics cannot directly pass through the cell membrane in passive targeting under normal physiological conditions, unless the nanoplastics can fit in the pores on the surface. For example, nanoplastics with a size smaller than the chorion pore of zebrafish embryos can penetrate the cell membrane and be translocated to the whole body [61]. Similarly, nanoparticles can cross the membrane of some cancer cells due to the enlarged size of the pores on the cell surface [62]. Passive transport mode can block entry of nanoplastics with comparatively large particle sizes and loosely allow entry of relatively smaller particles (Figure 3a).

Active targeting is a more complicated process compared with the passive mode due to various factors, such as size, shape, surface modification, “corona” compounds, and cell types. In addition, “corona” compounds are susceptible to the influence of other factors [57]. For example, smaller polystyrene beads are demonstrated to be internalized in greater numbers on the phagocytic uptake of macrophages [63]. Worm-like polyethyleneimine micelles covered by polypeptide “corona” can induce an increased cellular internalization by K562 cells with αvβ3 integrin overexpression compared with spherical micelles [56]. Studies report that murine RAW264.7 macrophages are more effective in translocation of carboxyl-modified nanopolystyrene compared with human endothelial HCMEC [64]. This implies that active targeting is dependent on the nanoplastics being internalized and studies should further explore it (Figure 3b).

#### 4.2.2. Nanoplastics Destroy Cell Membrane Structure Leading to Cell Death

Severe effects during the process of biological interactions include damage of membrane structure and cell death. Various factors, such as type and surface charge, play an important role in this process. For example, polyethylene nanoparticles fuse with the hydrophobic core of lipid bilayers and further form a network of disentangled, single polymeric chains. These complexes promote damage on the membrane structure and fluidity, and ultimately cell death [65,66]. Polystyrene particles with amino modifications extensively interact with cell membranes, resulting in dysregulated ion transport, signal transduction, membrane integrity, and even cell death [66,67,68,69] (Figure 3c).

## 5. Target Organelle Toxicity Induced by Nanoplastics

In vivo and in vitro experiments indicate that nanoplastics penetrate cell membranes and are internalized into cells, inducing intracellular biological effects [47,70,71,72]. Mitochondria, endoplasmic reticulum, and lysosome play vital roles in response to nanoplastics toxicity [70,73]. The subsequent sections present the main functions of the three organelles in eukaryotic cells in response to nanoplastic exposure.

### 5.1. Role of Mitochondria in Response to Nanoplastic Toxicity

Mitochondria is the major site for cell energy supply and oxidative phosphorylation. Exposure of mitochondria to external stimuli, such as nanomaterials, affects its normal structure and function, leading to metabolic and functional disorders [74]. Findings from previous studies indicate that internalized nanoparticles, including nanoplastics, are targeted to the mitochondria [70,75].

A previous study explored human bronchial epithelial BEAS-2B cells exposed to nanoplastics, and the results showed no significant morphological changes, such as swollen mitochondria. However, significant functional changes, such as abnormal energy metabolism, were observed in the mitochondria and the specific performances [32]. Organisms have a self-protection mechanism (enhanced autophagy) of oxidative mitochondrial activity that occurs to supply enough energy for regular homeostasis [71]. A study using zebrafish as the model animal showed that nanoplastics alter mitochondrial function by increasing oxygen consumption (OCR) from five aspects (rate basal, maximum, nonmitochondrial, basal mitochondrial, and mitochondrial spare capacity) in female gonad cells [47]. In addition, cells of Sterechinus neumayeri initiate a crucial self-protection mechanism of oxidative mitochondrial activity mediated by upregulation of superoxide dismutase (SOD), catalase (CAT), and metallothionein (MT) expression to maintain permeabilization of the mitochondrial membrane and activation of anti-apoptotic signaling of Bcl-2-caspase-8 after exposure to nanoplastics [76]. Moreover, a recent study reported the role of an anti-apoptotic-signaling cascade (Bcl2-Apaf1-caspase3) in response to nanoplastics using the *C. elegans* model. The study explored the upstream-signaling cascade of DNA damage (HUS1/Tel2p-p53-BH3), which exhibited the important self-protection strategy of the mitochondria in the regulation of nanoplastics toxicity [68]. These findings indicate that the mitochondria exhibit defensive mechanisms in response to nanomaterials toxicity, especially toxicity from nanoplastics (Figure 4a).

### 5.2. Role of Endoplasmic Reticulum in Response to Nanoplastics Toxicity

The endoplasmic reticulum is a subcellular organelle widely distributed in the cytoplasm of almost all eukaryotic cells. It is an important site for protein and lipid synthesis and plays a key role in intracellular signal transduction implicated in cell survival and apoptosis [77,78,79]. Studies have not fully explored whether nanoplastic particles penetrate the endoplasmic reticulum.

The imbalance of endoplasmic reticulum homeostasis occurs when organisms are under physiological or pathological stimulation. This imbalance leads to the accumulation of unfolded or misfolded proteins or changes in Ca^2+^ concentration in the endoplasmic reticulum lumen and ultimately induces unfolded protein response [77,80]. Significant upregulation of Grp78 and Grp170 expression is observed after exposure of coelomocytes to nanoplastics, indicating that exposure to nanoplastics induces pathways for oxidative stress alleviation and stress-related autophagy in endoplasmic reticulum [76]. Long-term exposure to nanoplastics at low doses causes endoplasmic reticulum stress, unfolded protein response, and fat metabolism disorder in intestinal cells of C. elegans. These effects are modulated through activation and phosphorylation of intracellular mitogen-activated protein kinase 14 (MAPK14), resulting in the upregulation of X-box binding protein 1 (XBP1). These proteins induce the endoplasmic reticulum unfolded protein response and dysregulation of sterol regulatory element-binding transcription factor 2 (SREBF2) and mediator complex subunit 15 (MED15). Subsequently, dysregulation of these factors affects lipid accumulation and modulate stearoyl-CoA desaturase (SCD) and stearoyl-CoA desaturase 5 (SCD5), ultimately inducing an innate immune response [71,81]. In addition, expression activating transcription factor 6 (ATF6), DDIT3 (DNA damage-inducible transcript 3 protein) and ERN1 (endoplasmic reticulum to nucleus signaling 1) is upregulated, inducing expression of immunofluorescence assay of microtubule-associated protein 1 light chain 3 (LC3-II) and accumulation of autophagosomes in bronchial epithelial BEAS-2B cells after exposure to nanoplastics. These changes indicate a potential autophagy regulation mechanism through the ER stress caused by misfolded protein aggregation [70]. These findings indicate that the endoplasmic reticulum is a crucial subcellular structure in response to the biological effects of nanoplastics (Figure 4b).

### 5.3. Role of Lysosome in Response to Nanoplastics Toxicity

The lysosome is an intracellular digestive organelle and the site for enzyme activities involved in removal of pathological cellular waste. Lysosomes can fuse with autophagosomes to form auto-phagolysosome in which lysosomal proteases degrade engulfed components [15,82,83].

Previous studies reported that nanoplastics internalized in eukaryotic cells accumulate in lysosomes. The accumulation of induced changes in lysosomal PH and membrane integrity ultimately cause lysosomal dysfunction [23,84]. Moreover, accumulation of nanoplastics induce the autophagic response through the activation of transcription factor, EB (TFEB), which further promotes an increase in lysosome–autophagosome fusion and, ultimately, enhances clearance of autophagic cargo [82]. Notably, overall blockage of autophagic flux if not alleviated can ultimately result in cell death due to severe damage of lysosomes [23]. A previous study reported that nanoplastics are translocated into lysosomes through a self-protection mechanism called the ‘Trojan horse’ effect. In this case, nanoplastics are covered by a layer of either soft or hard corona under a biological microenvironment, and intracellular membrane damage occurs once the surface of the corona is degraded [85]. Studies should further evaluate the effects of nanoplastics in lysosomes (Figure 4c).

## 6. Challenges and Future Research

The findings summarized in this review indicate that further studies should explore sources and effects of microplastics and nanoplastics. A summary of the aspects that should be evaluated is provided below.

### 6.1. Detection of Nanoplastics in the Environment

Although we have lessons from engineered nanomaterial research, existing analytical techniques are not still sufficiently developed to quantify nanoplastics in the environment, especially in biological samples. Previous studies report some methods used for the detection of nanoplastics, such as asymmetric flow field-flow fractionation coupled to multi-angle light scattering, fluorescent labeled, and Raman tweezers [86,87,88,89], but some issues still remain to be addressed. For example, fluorescent labeled, as the most frequently used detection method, particularly in cellular bioaccumulation, often involves artifacts, leading to false positives [88,90]. Considering that dye leakage and cellular autofluorescence might be the main sources of artifacts, dye core-wrapped and blank negative control should be used to alleviate the problem [88]. Given the discussion above, more efficient, convenient, and accurate analysis methods should be developed and applied to identify nanoplastics in the environment. Meanwhile, a complete set of detection systems for microplastics and nanoplastics in different media should be established to alleviate exposure of human to these plastic particles.

### 6.2. Elimination or Reduction of Microplastic or Nanoplastic Pollution

The separation and collection of nanoplastics from the environment is a challenge; however, there are some potential methods for reducing the levels of nanoplastics.

#### 6.2.1. Recycling

Approximately 6300 Mt of plastic waste was generated in 2015 [91]. More than 90% (5733 Mt) of plastic waste produced in 2015 was not recycled and were directly or indirectly released to the environment, and the level is projected to be 12,000 in 2050 [92]. Plastic waste released to the environment may eventually be degraded to microsize or nanosize; thus, recycling of plastic waste is an effective way to eliminate or reduce micropollutions or nanopollutions.

#### 6.2.2. Substitute Materials

Two types of materials can be used as substitutes for plastics. Chitin is a bioactive polymer widely used in industrial and biomedical fields. It is one of the most abundant natural polysaccharides [93]. Chitin has unique properties, such as high antibiosis activity, non-toxicity, ease of chemical synthesis and modification, and high biodegradability; therefore, it is a feasible substitute material for plastics [94]. Hemp fiber is a biodegradable polymer material widely used in the manufacture of ropes, automobile parts, polystyrene, and elastic building materials [95,96]. Hemp fiber is biodegradable, recyclable, and nontoxic; thus, it is a potential substitute for plastics.

#### 6.2.3. Degradation of Microplastic or Nanoplastic Pollutions

Degradation of environmental pollutants, including microplastics and nanoplastics, is conducted using physical, chemical, or biological methods. Previous studies show that chemical and physical methods are used to eliminate or reduce micropollutions or nanopollutions; however, these methods lead to the production of new pollutants or are associated with incomplete degradation [97,98]. Biodegradation can be applied to overcome the limitations of the traditional methods of degradation of pollutants [99]. Biodegradation is highly effective and has less side effects, thus playing a vital role in elimination or reduction of microplastic or nanoplastic pollutions from the environment.

### 6.3. Comprehensive Analysis of Nanoplastics Toxicity

#### 6.3.1. Toxicity of Aged Nanoplastics and Their Leachings

Nanoplastics released to the environment can result in absorption and leaching of environmental chemicals during their transport and transformation in different media [88,100]. Aged nanoplastics release high amounts of additives into the environment. In addition, their properties can be altered, increasing the potential toxicity. However, the effects of nanoplastics to ecology and humans have not been fully elucidated. Therefore, studies should explore the chemicals released from nanoplastics to the environment and their potential effects.

#### 6.3.2. Toxicity of Nanoplastics at Environmentally Relevant Concentrations (ERC)

Most studies expose nanoplastics to some model organisms at concentrations unlikely to exist in the real environment [61,100]. Few studies have explored the toxicity of nanoplastics at environmentally relevant concentrations (ERC). Therefore, studies should explore the effects of nanoplastics at the concentrations that they exist in the environment.

Several animal models have been used to explore various toxicity types induced by nanoplastics, such as reproductive toxicity, neuronal toxicity, and developmental toxicity. However, studies have not fully explored the potential toxicity of nanoplastics on humans [68,101,102]. Ineluctable exposure of nanoplastics to humans further drive the need to explore the effects of these particles on humans.

## 7. Conclusions

In this review, uncommon sources of nanoplastics, such as tire wear and laundry wastewater, were summarized in the present study to evaluate the relationship between nanoplastics and human health. The findings indicate that the potential sources of clothing and tire wear may result in high amounts of microplastics or nanoplastics in the environment. Further, the potential exposure routes, such as oral, inhaled, or dermal exposure, and the long-term biological effects of nanoplastics, such as crossing biological barriers and generation-crossing, were explored. In addition, the biointerface of nanoplastics was evaluated and the latent paths of entry into eukaryotic cells, including passive targeting and active targeting, were summarized. Furthermore, the effects of nanoplastic particles on intracellular target organelles, with mitochondria, endoplasmic reticulum, and lysosome as examples, were explored to describe the role of organelles in response to nanoplastic toxicity. These findings provide information on exposure of nanoplastics and the potential biological effects.

## Figures and Tables

**Figure 1 nanomaterials-12-01298-f001:**
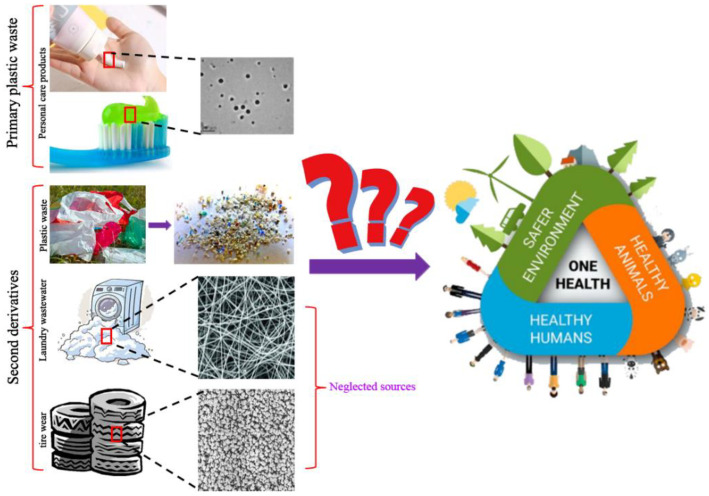
Another two sources of micro(nano)plastics in the environment.

**Figure 2 nanomaterials-12-01298-f002:**
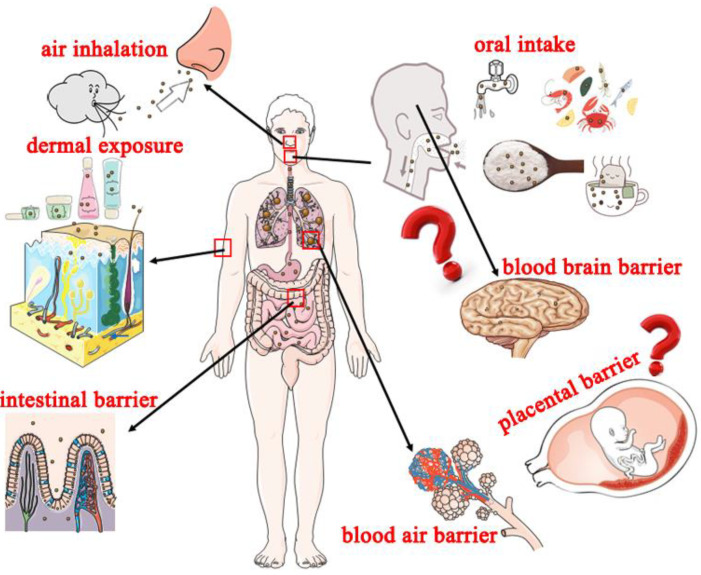
Potential exposure routes and adverse effects of nanoplastics on humans.

**Figure 3 nanomaterials-12-01298-f003:**
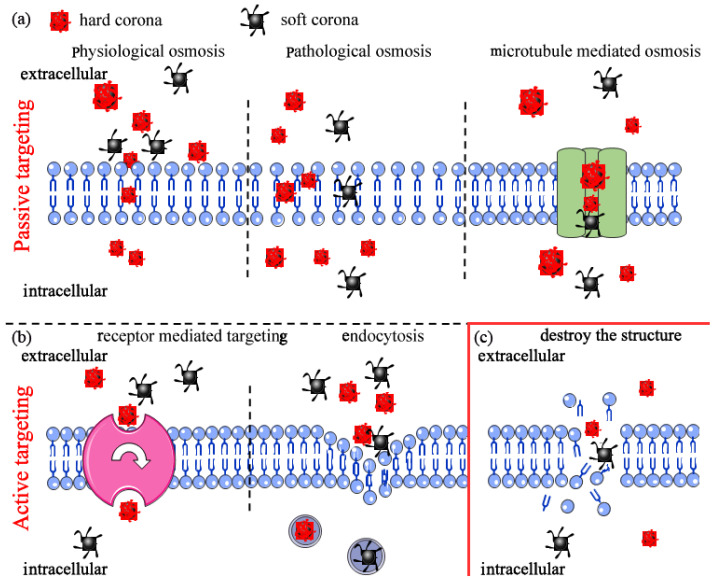
Biological effects of nanoplastics after cell internalization: (**a**) passive targeting; (**b**) active targeting; (**c**) cell death.

**Figure 4 nanomaterials-12-01298-f004:**
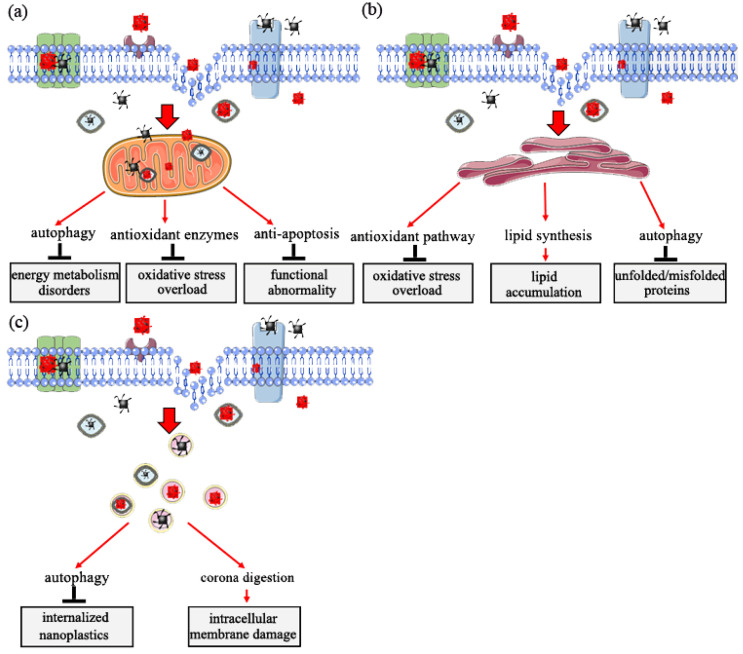
Target organelle toxicity induced by nanoplastics: (**a**) role of mitochondria in response to nanoplastics toxicity; (**b**) role of endoplasmic reticulum in response to nanoplastics toxicity; (**c**) role of lysosome in response to nanoplastics toxicity.

## Data Availability

No new data were created or analyzed in this review.

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
