# Peer review of "Nanoplastics and Human Health: Hazard Identification and Biointerface"

_nanomaterials, 2022, doi:10.3390/nano12081298_

Round 1

Reviewer 1 Report

The authors presented an overview of two underestimated sources of nanoplastics, exposure routes to illustrate hazard identification of nanoplastics, cell internalization and effects on intracellular target organelles. This paper shows many results from the available literature. There are several comments to be solved by the authors, after that I can recommend its publication in Nanomaterials.

  • The word “bionanoeffects” in line 28 is a little bit confusing. I suggest to be replaced.
  • The word “elucidated” in line 37 is not proper. I suggest to be replaced by “will be presented”.
  • The word “interactions” in line 141 is not proper. I suggest to be replaced by “behaviour”.
  • In the case of sub-chapter 4.2.1 the authors discussed mainly about the studies on nanoparticles and not on nanoplastics. More reports on nanoplastics have to be included. The same comment for sub-chapter 4.2.2.
  • The word “Summary” in line 278 is not proper. I suggest to be replaced by “Challenges and future research”.
  • I don’t understand why chapter 3. Comprehensive analysis of nanoplastics toxicity is after chapter 6. You have to move this part or to correct. However, there no relevant information for the review between lines 317 and 336.
  • The conclusion should be clearly marked.

Reviewer 2 Report

This paper provides a valuable summary of the human toxicology data on nanoplastics. I suggest for publication after revisions are made.

First, it is debatable that nanoplastic release from clothing and tire wear is “underestimated.” These are both well known sources of microplastics so it could be inferred that they would also be sources of nanoplastics. The bigger challenge is that analytical techniques are not sufficiently developed to quantify these sources of release (https://www.sciencedirect.com/science/article/pii/S1748013221002218).

Second, it is well known that engineered nanomaterials can cause artifacts in assays, particular cellular assays (e.g., https://pubs.acs.org/doi/abs/10.1021/acs.chemrestox.9b00165; https://www.tandfonline.com/doi/full/10.3109/17435390.2013.829590). This is also a key issue when testing nanoplastics. The authors should describe how insights from testing engineered nanomaterials can be applied to nanoplastics. Also, the authors should consider the work that has been done on improving the reliability of microplastic ecotoxicity measurements, as many of these considerations will also be relevant for nanoplastic human health testing (https://pubs.acs.org/doi/abs/10.1021/acs.est.0c03057). Lastly, the authors need to carefully review the following papers related artifacts during bioaccumulation measurements using fluorescent probes (https://www.sciencedirect.com/science/article/pii/S0048969719311829; https://setac.onlinelibrary.wiley.com/doi/full/10.1002/etc.4436). This possibility should be described in the paper, and the papers cited critically evaluated to see if this artifact could impact their results. Many of the papers cited suggesting bioaccumulation cited papers using fluorescent probes. Overall, perhaps it will be necessary for the authors to add a paragraph or short section on this topic.

Minor comment

Lines 192-194 – These positively charged particles have also shown cytotoxicity in studies (e.g., https://www.altex.org/index.php/altex/article/view/51; https://pubs.acs.org/doi/10.1021/bc060077y)

Round 2

Reviewer 1 Report

The authors have addressed all comments and I recommend the paper for publication.